# Novel Synthetic Opioids: The Pathologist’s Point of View

**DOI:** 10.3390/brainsci8090170

**Published:** 2018-09-02

**Authors:** Paolo Frisoni, Erica Bacchio, Sabrine Bilel, Anna Talarico, Rosa Maria Gaudio, Mario Barbieri, Margherita Neri, Matteo Marti

**Affiliations:** 1Department of Medical Sciences, University of Ferrara, 44121 Ferrara, Italy; paolo.frisoni@unife.it (P.F.); erica.bacchio@unife.it (E.B.); rosamaria.gaudio@unife.it (R.M.G.); mario.barbieri@unife.it (M.B.); 2Department of Life Sciences and Biotechnology (SVeB), University of Ferrara, 44121 Ferrara, Italy; bllsrn@unife.it; 3Department of Chemical and Pharmaceutical Sciences, University of Ferrara, 44121 Ferrara, Italy; anna.talarico@unife.it; 4Department of Morphology, Surgery and Experimental Medicine, Section of Legal Medicine, University of Ferrara, 44121 Ferrara, Italy; matteo.marti@unife.it; 5Collaborative Center for the Italian National Early Warning System, Department of Anti-Drug Policies, Presidency of the Council of Ministers, 00184 Roma, Italy

**Keywords:** fentanyl, NPS, synthetic opioids, MT-45, AH-7921, U-47700, forensic pathology

## Abstract

Background: New Psychoactive Substances (NPS) constitute a broad range of hundreds of natural and synthetic drugs, including synthetic opioids, synthetic cannabinoids, synthetic cathinones, and other NPS classes, which were not controlled from 1961 to 1971 by the United Nations drug control conventions. Among these, synthetic opioids represent a major threat to public health. Methods: A literature search was carried out using public databases (such as PubMed, Google Scholar, and Scopus) to survey fentanyl-, fentanyl analogs-, and other synthetic opioid-related deaths. Keywords including “fentanyl”, “fentanyl analogs”, “death”, “overdose”, “intoxication”, “synthetic opioids”, “Novel Psychoactive Substances”, “MT-45”, “AH-7921”, and “U-47700” were used for the inquiry. Results: From our literature examination, we inferred the frequent implication of fentanyls and synthetic opioids in side effects, which primarily affected the central nervous system and the cardiovascular and pulmonary systems. The data showed a great variety of substances and lethal concentrations. Multidrug-related deaths appeared very common, in most reported cases. Conclusions: The investigation of the contribution of novel synthetic opioid intoxication to death should be based on a multidisciplinary approach aimed at framing each case and directing the investigation towards targeted toxicological analyses.

## 1. Introduction

New Psychoactive Substances (NPS) are a heterogeneous class of non-controlled substances available on the global illicit drug market (e.g., smart shops, internet, “darknet”). The use of NPS, often consumed along with other drugs of abuse and alcohol, has resulted in a significantly growing number of emergency admissions due to overdoses and a high number of deaths. By July 2017, 739 different NPS were reported to United Nations Office on Drugs and Crime (UNODC) [1]. According to the The European Monitoring Centre for Drugs and Drug Addiction (EMCDDA)report, with an overall total of 38 substances, synthetic opioids have become the fourth largest group of substances monitored in 2017, after synthetic cannabinoids (179 substances), cathinones (130), and phenethylamines (94) [2].

The class of synthetic opioids include fentanyl, its analogs used in medical therapy (e.g., sufentanil, alfentanil, and remifentanil) [3], and novel non-pharmaceutical fentanyls not approved for human medical use (e.g., acetylfentanyl, acryloylfentanyl, carfentanil, α-methylfentanyl, 3-methylfentanyl, furanylfentanyl, 4-fluorobutyrylfentanyl, 4-methoxybutyrylfentanyl, 4-chloroisobutyrylfentanyl, 4-fluoroisobutyrylfentanyl, tetrahydrofuranylfentanyl, valerylfentanyl, cyclopentylfentanyl, and ocfentanil), and compounds with different chemical structures, such as MT-45 (1-cyclohexyl-4-(1,2-diphenylethyl)piperazine), AH-7921 (3,4-dichloro-*N*-{[1(dimethylamino)cyclohexyl]methyl} benzamide) and U-47700 (3,4-dichloro-*N*-[(1R,2R)-2-(dimethylamino)cyclohexyl]-*N*-methylbenzamide) [4,5]. They are used on their own or more often in combination with heroin or other opioids [6,7]. This paper critically examines the literature on deaths related to fentanyls and synthetic opioid overdose, alone or in combination with other psychoactive drugs (i.e., cocaine, benzodiazepine, alcohol, and other opioids) and investigates the characteristics and complexity of such deaths, analyzing all data useful for the forensic pathologist.

## 2. Materials and Methods

A literature search was carried out using public databases (such as PubMed, Google Scholar, and Scopus) to revise fentanyl-, fentanyl analogs- and others synthetic opioids-related deaths. Keywords including “fentanyl”, “fentanyl analogs”, “death”, “overdose”, “intoxication”, “synthetic opioids”, “Novel Psychoactive Substances”, “MT-45”, “AH-7921”, and “U-47700” were used for the inquiry. The data were collected from 1990 to June 2018. Only deaths were considered. There were no language restrictions. All types of papers were included. We also reviewed the reference lists of the identified publications and PubMed suggestions. The full texts of all the eligible papers were obtained. Finally, 128 articles were included in this review. The examined data included the circumstances of death (e.g., trauma, external injuries) and drug exposure (pharmaceutical versus illicit drug use). With regard to the concentrations of the compounds, blood and, when stated, liver, urine, stomach content, kidney, brain, vitreous humor, and nasal swab concentrations were reported. 

## 3. Results

### 3.1. Synthetic Opioid Overview

Novel synthetic opioids, similar to the classical opioids morphine and heroin, selectively bind to the µ-, δ-, and κ-opioid receptors in the peripheral and central nervous system (CNS), thereby simulating the effects of endogenous opiates. However, they generally show greater selectivity towards the µ-opioid receptor subtype than morphine [8]. Stimulation of the µ-opioid receptor promotes the exchange of GTP (Guanosine Triphosphate) for GDP (Guanosine Diphosphate) in the G-protein complex and subsequently inhibits adenylate cyclase in cells causing a decrease in intracellular cAMP (Cyclic Adenosine Monophosphate). In addition, the activation of the µ-opioid receptors inhibits calcium and potassium ion channel conductance [8]. All these molecular events cause cellular membrane hyperpolarization and inhibit tonic neural activity with a consequent reduction in the release of several neurotransmitters, such as substance P, GABA, dopamine, acetylcholine, and noradrenaline [9]. 

These neurochemical changes are mainly responsible for the pharmaco-toxicological effects induced by synthetic opioids. Typically, acute intoxication induced by “classical” and novel synthetic opioids is characterized by miosis (but later the pupils may become dilated), a reduced level of consciousness (CNS (central nervous system) depression), respiratory depression, hypoxia, acidosis, hypotension, bradycardia, shock, gastric hypomotility, paralytic ileus, pulmonary edema, lethargy, coma, and even death.

In the last few years, the novel fentanyls have become a serious concern. These substances currently dominate the synthetic opioid group, with a total of 28 reports since they first surfaced in 2012. Fentanyl, which is the prototypical compound of this class, is a synthetic, lipophilic phenylpiperidine opioid agonist. It was developed in the 1960s by Paul Janssen in Belgium, and it is now available therapeutically as an intravenous, transbucal, or transdermal preparation, commonly used for surgical anesthesia and to treat severe chronic pain [10]. Because of its highly potent opioid euphoric effects, fentanyl leads addicts to rapidly abuse this drug through a variety of different methods, including the oral abuse of transdermal fentanyl patches [4]. The clinical effects are dose dependent, ranging from the induction of analgesia alone by serum concentrations of 0.3–0.7 ng/mL to the loss of protective airway reflexes and CNS depression by serum concentrations >3 ng/mL [11]. In addition, fentanyl and its analogs produce drowsiness and euphoria [12]. The most common side effects may include nausea, dizziness, vomiting, fatigue, headache, and constipation. The repeated use of fentanyls leads to the development of tolerance and dependence [13]. Typical withdrawal symptoms involve sweating, anxiety, diarrhea, bone pain, abdominal cramps, and shivers or goose “flesh” [12,14]. 

From a pharmacological point of view, fentanyl and its analogs are significantly more potent than morphine, with effects’ magnitudes ranging from 1.5–7 times (butyrylfentanyl) to 10,000 times (carfentanil) those of morphine [14], and are characterized by high lipid solubility, rapid onset of action, and short duration of action. Like other types of opioid analgesics, such as morphine, methadone, and heroin, fentanyls produce their main effects by stimulating at nanomolar affinity the µ-opioid receptor [15,16,17,18]. In particular, they induce acute analgesia, relaxation, euphoria, sedation, bradycardia, hypothermia, depression of the central nervous system and respiratory function [16,19,20,21]. The last of the listed side effects poses the greatest danger to users and it is responsible for fentanyls-related significant morbidity and mortality [22]. The timely administration of the antidote naloxone [23] can rapidly reverse the severe respiratory depression caused by fentanyls [24], although multiple naloxone doses may be required [25]. The rapid administration of naloxone following fentanyl’s overuse is essential, because of the rapid onset of action of the drug that can cause respiratory depression within two minutes [26]. The optimal dose range and methods of administration of naloxone are still not clear. The UK Department of Health recommends the following naloxone dose regimen: an initial dose of 0.4 mg intravenously, followed by up to two doses of 0.8 mg. If the latter two doses are ineffective, a further 2 mg dose should be provided [27]. In fact, despite the fact that fentanyl shows an affinity for µ-opioid receptor (Ki ~ 1.346 nM) similar to that of morphine (Ki ~ 1.168 nM) [28], it displays a greater potency (EC_50_ ~ 0.15 nM) than morphine (EC_50_ ~ 2.4 nM) in functional biological assays [29]. On this basis, higher naloxone doses may be necessary to reverse the adverse effects of fentanyls (i.e., lofentanil) with greater µ-opioid receptor affinity (Ki ~ 0.023 nM) [30] and potency (EC50 ~ 0.03 nM) [29].

Because of the narrow therapeutic index of fentanyl (and, presumably, of its analogs) its recreational use is highly dangerous, especially in opioid-intolerant users. High doses might hasten death due to respiratory arrest and pulmonary edema. Fentanyl analogs are clandestinely developed for recreational use [15,30,31,32,33,34]. These compounds have been synthesized by modification or replacement of the fentanyl’s propionyl chain or by replacement of its ethylphenyl moiety. The obtained analogs have been further modified by substitution with fluoro, chloro, or methoxy groups at the *N*-phenyl ring. Among fentanyl analogs, carfentanil [methyl 1-(2-phenylethyl)-4-(*N*-propanoylanilino) piperidine-4-carboxylate] is considered one of the most lethal opioids, showing an extremely high clinical potency. It is used in research and, in some countries, as a veterinary medicine to immobilize large animals. Between November 2016 and April 2017, carfentanil was involved in at least 61 deaths in eight European countries. The vast majority of those deaths are related to heroin consumption [2]. 

Fentanyls have been found in a range of physical and dosage forms in Europe. The most common form is powder, but they have also been detected in liquids and tablets. E-liquids containing fentanyls that can be vaped using electronic cigarettes have also been reported [2]. 

These forms are easily absorbed through more convenient administration routes than injection, yet provide the consumers with psychoactive effects similar to those obtained with injectable forms. However, their use may pose a high risk of accidental overdose. In fact, nasal sprays and e-liquids could make fentanyls use more attractive and socially acceptable, promoting their spread and usage. 

In addition to fentanyls, other novel synthetic opioids with chemical structures different from fentanyl, i.e., MT-45, AH-7921, and U-47700, have appeared on the recreational drug market and are causing intoxication and potentially fatal outcomes in consumers. MT-45 (1-cyclohexyl-4-(1,2-diphenylethyl) piperazine)), also known as IC-6, CDEP, AC1L8SAC, and NSC 299236, has shown particular effects, such as paresthesia in limbs, hand weakness, balance disturbances, vision impairments, and hearing impairment or loss [35]. In three cases, unusual side effects have been reported, such as loss and depigmentation of hair, folliculitis and dermatitis, painful intertriginous dermatitis, and elevated liver enzymes [36]. MT-45 has been associated with many reports of fatal intoxications in Europe; in particular, Sweden reported 28 analytically confirmed deaths between November 2013 and July 2014 [37,38]. In vitro and in vivo metabolism studies have shown that MT-45 is biotransformed into active hydroxylated compounds [39] that may contribute to the overall pharmaco-toxicological profile of MT-45 in vivo [40]. AH-7921 (3,4-dichloro-*N*-[1(dimethylamino) cyclohexyl] methyl benzamide) belongs to a series of compounds known as cyclohexylamines [41]. The drug exhibited similar potency to morphine in preclinical studies [42]. The compound is taken orally, nasally, by smoking, and, less commonly, by intravenous injection [43]. The main clinical effects included hypertension, tachycardia, and seizures. The first death associated with AH-7921 use was reported by Norway in December 2012 [43]. There has been one confirmed fatality from AH-7921 in the United States, but a number of deaths have been associated with this drug in Europe [44,45].

U-47700 (3,4-dichloro-*N*-[(1R,2R)-2-(dimethylamino)yclohexyl]-*N*-methylbenzamide) is a structural isomer of AH-7921. It is also known as “fake morphine” or “U4” in the recreational drug market and it is sometimes also referred to as “pink”, because impurities in its synthesis cause the drug powder to be slightly pink in color. In preclinical studies, U-47700 is about 1/10 as potent as fentanyl, but 7.5 times more potent than morphine [46,47]. During 2016, a significant number of U-47700 acute intoxication cases were reported in the USA. The clinical symptoms are consistent with those of traditional opioids [5]. U-47700 has caused at least 46 deaths from overdose in the United States [48,49,50,51].

### 3.2. Circumstantial Data and External Examination

It is very important to sample any potential drug-containing material in the area around a dead body, taking into account the numerous opioid administration routes (drug paraphernalia, powders, syringes, vials, pills, patches etc.). It is also important to look for signs of administration on the body, considering, however, that about 20% [52] of subjects take fentanyl or analogs by inhalation, ingestion, or, rarely, transdermal route, therefore puncture marks are not always evident.

Even in the absence of external signs specific for opioid intoxication, it is possible to observe non-specific signs of asphyxia, such as petechiae [53].

### 3.3. Autopsy-Pathological Findings

The data from the literature review showed that the new synthetic opioids produce similar clinical effects as the traditional ones [54]; therefore, we sought the typical findings of heroin intoxication trying to capture the differences and identify additional typical findings of fentanyl- and synthetic opioid-related deaths. The autopsy findings collected from the case reports treated in the literature were homogeneous with respect to the detected findings. The routine histological data were not very specific and did not reveal indicative signs of intoxication [13,55].

#### 3.3.1. Central Nervous System

The major autoptic relief found in the CNS was cerebral oedema. This finding was reported for several drugs (fentanyl [56,57], acetylfentanyl [58], butyrylfentany [59], furanylfentanyl [60,61], ocfentanil [62], AH-7921 [63], U-47700 [64], MT-45 [65]). A case of fatal cerebral hemorrhage induced by acetylfentanyl was reported [66], and another case of a 19-month-old girl poisoned by a transdermal administration of fentanyl who developed leukoencephalopathy was described [67]; in this case, the girl survived, however, investigators declared that this is not always the outcome [55].

#### 3.3.2. Cardiovascular System

A reported uncommon intoxication symptom is chest pain mimicking acute coronary syndrome with non-specific T-wave changes on the electrocardiogram [68]. It is necessary to distinguish the alterations induced acutely by the drug from those due to pre-existing pathologies. Most of the observed cardiovascular pathological findings, such as hypertrophy [69], cardiomegaly [56,70,71], cardiac fibrosis [72,73], atherosclerosis [69,74], are not attributable to an acute intoxication but, in some cases, they may be compatible with chronic drug intake. The presence of pericardial petechiae [53] can be interpreted as a generic sign of asphyxia, due to opioid-related respiratory failure.

#### 3.3.3. Pulmonary

The main effect of fentanyl and its analogues on the respiratory system is respiratory depression. Furthermore, fentanyl can cause chest wall rigidity and apnea, particularly with rapid intravenous administration [75], a factor that can contribute to respiratory failure. Rare adverse effects after fentanyl usage include diffuse alveolar hemorrhage immediately after insufflating fentanyl powder [72]. The major pathological findings are pulmonary congestion [53,54,69,70,73,74,76,77,78,79,80,81,82] and pulmonary oedema, which are common to all the investigated drugs [57,58,70,73,74,78,79,80,81,83]. Signs found occasionally are petechiae on the pleura [57,82] and aspiration of gastric contents inside the trachea and bronchi [57,76,81].

A few cases of fentanyl patch aspiration have been reported, where the patch was found in the airways [71,84]. Microscopically, small amounts of foreign material have been reported in the lungs, consistent with prior intravenous drug abuse [80].

#### 3.3.4. Others

Another common sign is generalized visceral congestion [51,60,65]. Hepatic parenchyma alterations, such as liver cirrhosis [74], chronic active hepatitis [82], fatty degeneration [62,70,84], hepatomegaly [62,70], are common but due to pre-existing conditions or chronic abuse of narcotics.

### 3.4. Sampling

The samples commonly taken for toxicological analysis consisted of peripheral blood [84], central blood [85], urine, and liver. Less commonly collected samples were vitreous humour [78], brain, kidney, bile, and gastric content. 

Among these, the least susceptible site to post-mortem redistribution is the liver (in relation to fentanyl) [86]. However, there is currently no consensus on the ideal sampling site [87].

### 3.5. Lethal Concentrations

The lethal concentrations found in the literature are reported in Table 1. For each drug, the routes of administration and relative potency compared to morphine are shown, in addition to the dose (and the corresponding tissue). Table 2 shows the concentration data reported in multidrug-associated deaths.

## 4. Discussion

This review deals with human pathological findings that are directly attributable to the known toxic actions of fentanyls and other synthetic opioids. In the past few years, it has become more and more evident that fentanyls and other synthetic opioids are potentially extremely harmful. Fentanyl-related deaths have increased over the years [47,52], so it is necessary to review the data available on these analog NPS. The results from our literature analysis revealed the lethal potential of fentanyls and other synthetic opioids; a large number of different routes of substance administration have also surfaced, through which all of these compounds are potentially lethal.

In addition, a broad range of side effects associated with fentanyls and other synthetic opioids have emerged, posing serious health issues, which primarily concern the central nervous system, cardiovascular and pulmonary systems, and liver. Macroscopic examinations, autopsy data, and histopathological elements were collected from the literature, leaving evidence that mainly refers to opioid intoxication. The investigation of the cause of death provoked by fentanyl or other synthetic opioid abuse was based on a multidisciplinary approach aimed at framing each case and directing the investigations towards targeted toxicological analyses. This approach should be adopted in all cases of death from uncertain or questionable causes [66,70]. Past medical history and ante-mortem distribution, crime scene investigation, post-mortem toxicology examination, and toxicology findings should be carefully analyzed and considered on a case-by-case basis in light of all other data [11,98,105,106,107].

The examination of the literature showed that a large number of deaths associated with fentanyl and other synthetic opioids involved the abuse of other psychoactive substances. In a previous review [22] of various case studies of fentanyl-related deaths, it was speculated that the deaths were associated with drugs of abuse such other opiates (up to 64%), cocaine (up to 65%), cannabinoids (up to 50%), amphetamines (up to 40%), but also ethanol (up to 22.9%) and medicines like barbiturates (up to 27%), benzodiazepines (up to 52.2%), antidepressants (up to 48%). These data can lead to a series of considerations. The first is inherent in the contribution to the deaths of the substances detected together with synthetic opioids. Animal studies have shown that some classes of substances, such as benzodiazepines, have a synergistic effect with opioids.

Although the pharmacokinetic and pharmacodynamic interactions between benzodiazepines and opioids are not yet fully understood [108], pre-clinical studies suggest a synergistic effect on opioid-induced respiratory depression (measured as % increase in pCO_2_) [109]. Forensic data show the risks of this drug combination: the concomitant use of benzodiazepines and “traditional” opioids is associated with the occurrence of opioid overdoses [110,111]. This occurrence is very common, and the co-use of opioid and benzodiazepine could be aimed at amplifying the subjective effects of the opioid [112,113]. However, the current knowledge does not permit to establish the exact contribution of benzodiazepines in opioid-related deaths, considering that many opioid addicts are also chronic benzodiazepine users [114,115].

Regarding the co-administration of ethanol and opioids, it can be dangerous because it enhances the positive subjective effects that contribute to the abuse and affects physical and cognitive functions. It is no coincidence that alcohol and opioid abuses often coexist [116]. Fatal intoxications involving opioids are frequently associated with alcohol use and are likely due to combined CNS- and respiratory-depressant effects [117,118]. 

Concerning the co-administration of cocaine and synthetic opioids, animal studies have shown that combinations of cocaine and remifentanil can lead to a strong additivity [119], maybe for their synergistic action on the mesolimbic dopaminergic system [120]. Unfortunately, data on humans are not yet reported in the literature. Recently, there has been an increase in cocaine-related overdoses; however, a large part of this increase is due to the simultaneous intake of opioids, especially synthetic ones [121]. Often, the use of a synthetic opioid is accidental, due to an unknown contamination of a cocaine stock [122,123].

The same assumption can be formulated for the deaths from the co-administration of heroin and fentanyl. A recent study showed a strong association between the number of tested samples of seized drugs where fentanyl was detected and unintentional overdose deaths in which fentanyl was also identified [124]. This data are consistent with previous studies that have shown that a significant proportion of drug users unintentionally consume fentanyl, which is present in the substances they are taking [7,122,125]. However, the real diffusion of illicit fentanyl use in the general population is difficult to assess, because routine toxicology screens will not detect synthetic opioids that have little structural homology to morphine and other commonly tested opioids [14].

## 5. Conclusions

In Conclusion, the confirmation or exclusion of opioid overdoses is one of the major challenges for forensic pathologists, considering what has been said and that autopsy findings are not specific. It is therefore necessary that the forensic pathologist have a broader approach and, on the basis of the data collected, request a chemical-analytical analysis to point out NPS [87]. 

The role of the forensic pathologist in close collaboration with the forensic toxicologist is very important: together, they can identify new cases of fentanyl and synthetic opioid intoxication. The identification of NPS is essential to stop the social problems related to the spread of these new dangerous and highly addictive substances among our population.

## Figures and Tables

**Table 1 brainsci-08-00170-t001:** Lethal concentrations data reported in the literature.

	Potency Ratio to Morphine [14]	Administration Route Associated with Overdose	Blood Concentration (ng/mL)	Other Concentrations (Site, ng/mL)
Acetylfentanyl [53,58,80,88]	15.7	Nasal, intravenous	153–260 247.5–285 (heart)	Liver 100–2400 ng/g; urine 2.6–2720 ng/mL; stomach content 880 ng/mL; vitreous humor 131–240 ng/mL.
Alpha-Methylfentanyl [89]	56.9	Intravenous	3.1	liver 78 ng/g; bile 6.4 ng/mL.
Butyrylfentanyl [59,71]	1.5–7.0	Nasal, rectal, intravenous, sublingual	66–99 ng/mL; 39–220 ng/mL (heart)	liver 41–57 ng/g; kidney 160 ng/g, muscle 100 ng/g; vitreous humor 32 ng/mL; bile 260 ng/mL; urine 64 ng/mL; gastric contents 590 ng/mL; brain 93 ng/g.
Carfentanil [90]	10,000		0.11–0.88	
4-Fluorobutyrfentanyl [91]	Unknown	By smoking	91–112	urine, 200–414 ng/mL; liver, 411–902 ng/g; kidney 136–197 ng/g.
Furanylfentanyl [49,60]	Unknown	Nasal, intravenous	0.43–26	
3-Methylfentanyl [83,92]	48.5–7000	Intravenous	0.3–1.9	
Ocfentanil [62,93]	90	Nasal, by smoking	9.1–15.3; 23.3–27.9 (heart)	vitreous humor 12.5 ng/mL; urine 6.0 ng/mL; bile 13.7 ng/mL; liver 31.2 ng/g; kidney 51.2 ng/g; brain 37.9 ng/g; nasal swabs 2999 ng/swab.
AH-7921 [5,44,64,66]	Unknown	Oral, nasal, by smoking, intravenous	330–6600 480–3900 (heart)	urine 760–6000 ng/mL; bile 17,000 ng/mL; liver 530–26,000 ng/g; kidney 7200 ng/g; brain, 7700 ng/g; vitreous humor 190 ng/mL; stomach content, 40 μg/mL.
U-47700 [5,49,94,95]	7.5	Oral, nasal, intrarectal, smoking, intravenous	59–525 1347 (heart)	Urine 360–1393 ng/mL; liver 430–1700 ng/g; kidney 270 ng/g; lung 320 ng/g; brain 97 ng/g.
MT-45 [5,66,96]	Unknown	Oral, nasal, intrarectal, intravenous	520–660 1300 (heart)	Urine 370 ng/mL; vitreous humor 260 ng/mL; gastric content 49 μg/mL; liver 24 μg/g.
Fentanyl [70,82,86,87,97,98]	100	Oral, transdermal, nasal, intravenous	0.5–383	Urine 2.9–895 ng/mL; gastric content 31.6–745 μg/mL; liver 5.8–613 μg/g.

**Table 2 brainsci-08-00170-t002:** Concentrations data reported in multidrug-associated deaths.

	Associated Drugs	Blood Concentration (ng/mL)	Other Concentrations (Site, ng/mL)
Acetylfentanyl	Butyrylfentanyl [69,81]	Acetylfentanyl: 21–38; 32–95 (heart) Butyrylfentanyl: 3.7–58; 9.2–97 (heart)	Acetylfentanyl: vitreous humor 38–68 ng/mL; bile 330 ng/mL; urine 8–690 ng/mL; gastric contents 170–28,000 ng/mL; brain 200 ng/g; liver 110–160 ng/g. Butyrylfentanyl: vitreous humor 9.8–40 ng/mL; bile 49 ng/mL; urine 2–670 ng/mL; gastric contents 170–4000 ng/mL; brain 63 ng/g; liver 39–320 ng/g.
	Fentanyl [85,99]	Acetylfentanyl: 0.13–12 Fentanyl: 0.24–21	
	Fentanyl, heroin [100]	Acetylfentanyl: 12 Morphine (free): negative Morphine (total): 20 Fentanyl: 15	
	Fentanyl, heroin, cocaine [100]	Acetylfentanyl: 9 Morphine (free): 30 Morphine (total): 60 Fentanyl: 20 Cocaine: 70 Benzoylecgonine: 970	
	Fentanyl, heroin, alprazolam [100]	Acetylfentanyl: 2 Morphine (free): 20 Morphine (total) <20 Alprazolam 30 Fentanyl: 19	
	Furanylfentanyl, diphenhydramine [49]	Acetylfentanyl: 0.65 Furanylfentanyl: 12.9 Diphenhydramine: 140	
	Morphine [100]	Acetylfentanyl: 400 Morphine (free): 30 Morphine (total): 70	
	Alprazolam [100]	Acetyl Fentanyl 560–600 Alprazolam 20–230	
	4-MethoxyPV8 and others [53]	Acetylfentanyl: 153 4-MethoxyPV8: 389 7-Aminonitrazepam: 200 Phenobarbital: 7700 Methylphenidate: 30	Acetylfentanyl: urine 240 ng/mL; gastric contents 880 ng/mL. 4-MethoxyPV8: urine 245 ng/mL; gastric contents 500 ng/mL.
Butyrylfentanyl	U-47700 [49]	Butyrylfentanyl: 26 U-47700: 17	
Furanylfentanyl	U-47700 [49]	Furanylfentanyl: 2.5–26 U-47700: 105–490	
	U-47700, Heroin [49]	Furanylfentanyl: 56 U-47700: 107 Morphine: 48	
	Fentanyl [60]	Furanylfentanyl: 0.4 Fentanyl: 1.27	
	Carfentanil [101]	Furanylfentanyl: 0.34 Carfentanil: 1.3	
U-47700	Fentanyl [102]	U-47700: 13.8 Fentanyl: 10.9	U-47700: urine 71 ng/mL
	Diphenhydramine [49]	U-47700: 103 Diphenhydramine: 694	
	Diphenhydramine, alprazolam, doxylamine [96]	U-4770:190 Diphenhydramine: 140 Alprazolam: 120 Doxylamine: 300	
Fentanyl	Heroin [82,98,100]	Fentanyl: 2.7–16 Morphine (free): <20–100 Morphine (total): 30–240	
	Heroin, hydromorphone [100]	Fentanyl: 15 Morphine (free): 20 Morphine (total): 60 Hydromorphone (free:) <20 Hydromorphone (total): 40	
	Heroin, methamphetamine [100]	Fentanyl: 0.004 Morphine (free): 100 Morphine (total): 90 Methamphetamine: 270	
	Heroin, methadone, alprazolam [100]	Fentanyl 7–38 Morphine (free): 20–50 Morphine (total): 40–80 Methadone: 320–400 Alprazolam: 30	
	Oxycodone [97]	Fentanyl: 14 Oxycodone: 420	
	Oxycodone, citalopram [97]	Fentanyl: 6.7 Oxycodone: 500 Citalopram: 200	
	Oxycodone, codeine [97]	Fentanyl: 10 Oxycodone: 270 Codeine: 280	
	Methadone [103]	Fentanyl: 5 Methadone: 540 Oxycodone: 70 Trazodone: 246	
	Hydrocodone [103]	Fentanyl: 90 Hydrocodone: 240	
	Morphine [103]	Fentanyl: 10 Morphine (total): 3230	
	Cocaine [97,103]	Fentanyl: 12–34 Cocaine: 50–780 Benzoylecgonine: 31–4100	
	Methanphetamine [71]	Fentanyl: 8.6 Methanphetamine: 1456	
	Phenobarbital, nordiazepam, diazepam	Fentanyl: 20 Phenobarbital: 7000 Nordiazepam: 72 Diazepam: 58	
	Bromazepam [55]	Fentanyl: 60.6; 94.1 (heart) Norfentanyl: 19.8; 50.1 (heart) Bromazepam: 874	Fentanyl: urine 152.2 ng/mL; brain 70.4 ng/g; kidney 161.3 ng/g; stomach content 536.8 ng/mL; liver 203.2 ng/g; bile 274.2 ng/mL Norfentanyl: urine 172.2 ng/mL; brain 5.6 ng/g; kidney 172.9 ng/g; stomach content 54.4 ng/mL; liver 164.6 ng/g; bile 436.4 ng/mL
	Alprazolam, tramadol [103]	Fentanyl: 12 Alprazolam: 13 Tramadol: 1500	
	7-Aminoclonazepam, Sertraline [82]	Fentanyl: 13.8 7-Aminoclonazepam: 57.1 Sertraline: 91.9	
Tetrahydrofuranylfentanyl	U-49900, Methoxy-Phencyclidine [104]	Tetrahydrofuranylfentanyl: 339 U-49900: 1.5 Methoxy-Phencyclidine: 1.0	Tetrahydrofuranylfentanyl: urine >5000 ng/mL U-49900: urine 2.2 ng/mL Methoxy-Phencyclidine: 31.8 ng/mL

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
