# Peer review of "Novel Synthetic Opioids: The Pathologist’s Point of View"

_brainsci, 2018, doi:10.3390/brainsci8090170_

Round 1
Reviewer 1 Report
The manuscript has the potential for publication in a peer-reviewed journal once significant revision has been undertaken. The revised manuscript has numerous instances of poor grammar, inaccurate and imprecise statements. I recognize that the authors primary language is not English. However, the manuscript should have been scrupulously reviewed by someone well versed in English prior to submission.
It is the first I have reviewed in which the manuscript does not seem to have been closely scrutinized by a co-author who is well-versed in English. My original review included the following: "(x) Extensive editing of English language and style required." In examining the revised manuscript there are still numerous instances in which the wording is unacceptable.
Examples are numerous and include the following just in the Abstract:
Line 24: "a wide scenario of fentanyls"
Line 27: "According to paper examined multidrug related deaths are very common."
Line 28: "fentanyl or synthetic opioids intoxication" Fentanyl is a synthetic opioid, as was previously stated to the authors.
Line 28: misused semi-colon
I had hoped the authors would have gone back over the paper, line by line, with someone well-verse in scientific writing in English.
Author Response
As you kindly requested, after making the changes in the text requested also from the review 2 (highlighted, in yellow, in the manuscript), we asked to a native speaker, different from the previous, to perform an estensive revision of the English language. The changes are highlighted in the text with the "track changes" system of Word program.

Reviewer 2 Report
In my opinion the manuscript is interesting for broad specter of readers. The public health problem of illicit novel synthetic opioids is increasing in last few years in pandemic way and represents a dinamic challenge for researcher, clinicians and policy makers.
I would like to suggest slight change in title: Novel synthetic opioids: pathologist point of view. The main reason is that methadone, buprenorphine an others are also synthetic opioids but in this paper only novel synthetic opioids (according definition) are included.
According to UNODC, till middle of 2017, 702 of the cumulative global total of 739 different NPS reported to UNODC, had been reported by 41 European countries (please see United Nations Office on Drugs and Crime. (2017): 2017 Global Synthetic Drugs Assessment.)
Among new generation of novel synthetic opioids with chemical structures different from fentanyl as AH-7921, U-47700, MT-45 also W-18, W-15, U-50488, U-49900, U-51754…appeared on illicit novel synthetic market. How would you comment, did you find cases of death due to this substances in reviewed literature?
Author Response
Thanks for the precious suggestions, as requested, we have proceeded to modify the title of the article in “Novel synthetic opioids: pathologist point of view”, we have consulted the UNODC and inserted in the text. We have also looked for cases of death due to novel synthetic opioids with chemical structures different from fentanyl and we an article that we lacked:Krotulski, A.J., Papsun, D.M., Friscia. M., Swartz. J.L., Holsey, B.D., Logan, B.K. Fatality Following Ingestion of Tetrahydrofuranylfentanyl, U-49900 and Methoxy-Phencyclidine. J Anal Toxicol. 201842(3), e27-e32. DOI: 10.1093/jat/bkx092. The article has been included in the text and cited in table 2. All the changes made are highlighted in yellow in the manuscript.

Round 2
Reviewer 1 Report
Please see the attached manuscript with numerous suggested changes. Wording is frequently awkward and imprecise.

Author Response
Thank you, for the suggested changes, I proceeded to reply to all your comments in the text you attached. I have also made all the changes in the manuscript text that I will resubmit, the changes in the manuscript are highlighted in the text with the "track changes" system of Word program.
